# Efficacy of mesenchymal stromal cells in the treatment of unexplained recurrent spontaneous abortion in mice: An analytical and systematic review of meta-analyses

Xiaoxuan Zhao[1☯], Yijie Hu[2☯], Wenjun Xiao[2], Yiming Ma[2], Dan Shen[1], Yuepeng Jiang[2], Yi Shen[1], Suxia Wang[1], Jing Ma[1]*

1 Department of Traditional Chinese Medicine (TCM) Gynecology, Hangzhou TCM Hospital Affiliated to Zhejiang Chinese Medical University, Zhejiang Province, Hangzhou, 310007, China, 2 The Third Clinical Medical College, Zhejiang Chinese Medical University, Zhejiang Province, Hangzhou, 310053, China

☯ These authors contributed equally to this work.
* 2021b085@zcmu.edu.cn

**Data Availability Statement:** All relevant data are within the paper and its Supporting Information files.

## Abstract

### Objectives

Unexplained recurrent spontaneous abortion (URSA) remains an intractable reproductive dilemma due to the lack of understanding of the pathogenesis. This study aimed to evaluate the preclinical evidence for the mesenchymal stromal cell (MSC) treatment for URSA.

### Methods

A meticulous literature search was independently performed by two authors across the Cochrane Library, EMBASE, and PubMed databases from inception to April 9, 2023. Each study incorporated was assessed using the Systematic Review Centre for Laboratory Animal Experimentation (SYRCLE) risk of bias tool. The amalgamated standardized mean difference (SMD) accompanied by 95% confidence interval (CI) were deduced through a fixed-effects or random-effects model analysis.

### Results

A total of ten studies incorporating 140 mice were subjected to data analysis. The MSC treatment yielded a significant reduction in the abortion rate within the URSA model (OR = 0.23, 95%CI [0.17, 0.3], *P*<0.00001). Moreover, it elicited a positive modulatory impact on the expression profiles of several inflammatory cytokines in the decidual tissue of URSA murine models, inclusive of IL4 (SMD 1.63, 95% CI [0.39, 2.86], *P* = 0.01), IL10 (SMD 1.60, 95% CI [0.58, 2.61], *P* = 0.002), IFN-γ (SMD -1.66, 95%CI [-2.79, -0.52], *P* = 0.004), and TNF-α (SMD -1.98, 95% CI [-2.93, -1.04], *P*< 0.0001). Subgroup analyses underscored that the administration mode of intraperitoneal and uterine horn injections, and sources of bone MSCs and adipose-derived MSCs contributed positively to the expression of IL4, IL10, and

**Funding:** This study was funded by the National Natural Science Foundation of China (82305294), the TCM Science and Technology Project of Zhejiang Province (2023ZR038), the Research Project of the Affiliated Hospital of Zhejiang Chinese Medical University (2022FSYYZQ16) and Hangzhou Health Science and Technology Project (A20230675) to ZXX; the National Natural Science Foundation of China (82305299) and the TCM Science and Technology Project of Zhejiang Province (2022ZA120) to MJ; the China Postdoctoral Science Foundation (2023M733193), the TCM Science and the Research Project of Zhejiang Chinese Medical University (2022RCZXZK29) to JYP; the Key project of Zhejiang Provincial Administration of Traditional Chinese Medicine (2022ZZ025) to WSX. None of the authors received a salary from the funders, and all the authors listed have approved the enclosed manuscript.

**Competing interests:** The authors have declared that no competing interests exist.

decreased the expression of IFN-γ in decidual tissue of URSA ($P<0.05$). Conversely, the tail vein injections subgroup was observed with no statistical significance ($P>0.05$).

## Conclusions

The findings underscore the considerable potential of MSCs in URSA therapy. Nonetheless, the demand for enhanced transparency in research design and direct comparisons between various MSC sources and administration routes in URSA is paramount to engendering robust evidence that could pave the way for successful clinical translation.

## 1 Introduction

Recurrent spontaneous abortion (RSA) refers to the experience of two or more consecutive pregnancy losses with the same spouse prior to the 28th week of gestation [1,2], which affects approximately 1%-5% of the global female population. Although RSA is multi-etiological, half of the cases are inexplicably classified as unexplained recurrent spontaneous abortion (URSA) [3]. The currency of URSA can cause severe physical and psychological harm, encompassing continued anxiety, depression, placental disorders in future ongoing pregnancies, abdominal pain, and chronic endometritis, thus deserving close attention [4]. It is generally believed that approximately 80% of URSA cases are attributable to immune factors [5]. Consequently, immunotherapy has emerged as the primary strategy for addressing URSA. Existing research has suggested beneficial outcomes in URSA prognosis through the utilization of treatments such as paternal leukocytes, intravenous immunoglobulin (IVIg), and growth factors, such as granulocyte-colony stimulating factor. Nonetheless, these approaches do not yield satisfactory results in all instances and may induce adverse effects [6], necessitating new therapeutic methods that can restore the fetomaternal interface immune microenvironment.

Mesenchymal stem cells (MSCs) are mesoderm-derived stem cells. Owing to their inherent capacity for self-renewal and multidirectional differentiation [7], MSCs are increasingly recognized as potent tools within the realm of regenerative medicine. They could also demonstrate in vivo and in vitro immunoregulatory capacities, positioning them as promising agents for managing inflammatory and autoimmune diseases [8]. It has been verified that MSCs could suppress the proliferation of Th1 cells, natural killer (NK) cells [9], modulate the function of dendritic cells (DCs) [10], and induce the expansion of regulatory T cells in abortion-prone mice [11]. These cells modulate the immunoregulatory functions at the maternal-fetal interface during pregnancy via the upregulation or induction of various immunoregulatory agents [12,13]. Given the robust role of MSCs in mitigating undesirable immune responses in the URSA murine model, we conjecture that MSC therapy could potentially enhance pregnancy outcomes in URSA. However, at this juncture, MSC treatment for URSA is confined to the preclinical stage [14], with the overall therapeutic efficacy yet to be definitively established. As such, several unresolved issues preclude the transition of MSC treatment for URSA into clinical practice, which include the need for rigorous confirmation of MSC efficacy and exploration of the optimal transplantation route. Fortunately, the accumulation of studies in this domain now enables a comprehensive review of cell-based treatment for URSA experimental models, thus providing a deeper understanding of the potential effects of treatment-related factors on pregnancy outcomes in URSA.

Thus, we have embarked upon a systematic review and meta-analysis to evaluate the efficacy of MSCs in the URSA murine model, with the intent of facilitating the application of

MSC treatment in clinical practice. The present meta-analysis may offer valuable insights for the design of randomized controlled clinical trials involving human subjects, as well as provide crucial data for future translational research.

## 2 Materials and methods

This study was conducted in accordance with the format given by the Preferred Reporting Items for Systematic Reviews and Meta-analysis (PRISMA) statement. We used the PRISMA checklist for the manuscript S1 Table [15]. The protocol did not exist in any database.

### 2.1 Eligibility criteria

**2.1.1 Animal models.** URSA murine models that were mated in female CBA/J mice with male DBA/2 were included. Other kinds of animal models were excluded, such as rats, rabbits, pigs, monkeys and dog models.

**2.1.2 Intervention.** For inclusion, studies must have used MSC treatment. Homologous MSC sources from all tissues were incorporated, and all routes, dosages, timing, and frequency of administration were included. Exclusion of stem cells that are combined with other interventions.

**2.1.3 Comparator.** All comparators were taken into account, including but not limited to placebo and vehicle control.

**2.1.4 Outcome.** The primary outcome is the embryo absorption rate, and the secondary outcome reported in the articles should have included one or more cytokines, such as the level of IL4, IL10, IFN-γ, or TNF-α.

**2.1.5 Study design.** Randomized controlled experimental studies published in English were included. And in vitro experiments, clinical trials, review articles, abstracts, editorials, commentaries, and letters were not included, along with unpublished grey literature.

### 2.2 Search strategy

Two authors (WJX, YJH) independently searched for all published English-language studies through PubMed, EMBASE and Cochrane Library. The search scope was from database construction to April 9, 2023. We used free words and MeSH words, and the search strategy was as follows: (RSA OR recurrent spontaneous abortion OR unexplained recurrent spontaneous abortion OR Recurrent pregnancy loss OR recurrent miscarriage) AND (mesenchymal stem cells OR MSCs). Further details of the search strategy were shown in S2 Table.

### 2.3 Study selection and data extraction

Two researchers independently screened the article titles and abstracts, irrelevant citations were excluded only when both investigators agreed. All qualified articles were retrieved for full review, in which the two investigators evaluated the articles independently using predetermined inclusion and exclusion criteria. Disagreements or uncertainties were discussed by the two investigators and resolved by a third investigator when necessary.

Data was extracted from the collected literature, including first author, year, country of study, animal model, age of animals, number of animals per group, type of MSCs, the way of administration, time and frequency of injection, treatment duration, and outcome measures. For articles with missing data or the original data were not provided. Engauge Digitizer [16] was used to obtain the data by measuring charts, and we also emailed the corresponding authors. The SEM was converted to SD ($SD = \sqrt{n} \times SEM$).

### 2.4 Risk of bias assessment

The risk of bias was evaluated by two independent reviewers (YJH, WJX) using the systematic review center for laboratory animal experimentation (SYRCLE) risk of bias tool [17], and the SYRCLE tool features 10 different parameters including randomization, blinding, outcome reporting, and so on, each of the parameters was scored as "low risk", "high risk" or "unclear risk". Any disputes encountered during the evaluation process were resolved through discussion.

### 2.5 Statistical analysis

**2.5.1 Summary measures.** RevMan 5.3 and Stata 16 were utilized to analyze and integrate the data. All studies used standardized mean difference (SMD) as the the effect analysis statistic and its 95% confidence interval (CI) was provided. Dichotomous variables were expressed as Odds ratio (OR) values. Data following a normal distribution were presented in the form of mean ± SD. We converted medians and interquartile ranges to means and standard deviations according to the formula [18] for subsequent analysis.

**2.5.2 Heterogeneity analysis.** The $I^2$ test was used to evaluate the heterogeneity among studies. When the heterogeneity was low ($I^2$ <50%), the fixed-effects model was the first choice. If high heterogeneity was found ($I^2$>50%), the random-effects model was applied [19]. Furthermore, subgroup analysis was performed in the presence of significant heterogeneity.

**2.5.3 Risk of publication bias.** Publication bias in the studies was examined using the funnel plot and Egger's tests. RevMan 5.3 and Stata 16 were used to compute the meta-analysis and examined publication bias.

**2.5.4 Subgroup analysis.** To explore the source of heterogeneity and to identify the way of MSC administration on URSA, a subgroup analysis was performed. The factors contributing to high heterogeneity were determined based on the $I^2$ statistic.

## 3 Result

### 3.1 Search results

The selection process of the electronic databases were shown in the flow diagram (Fig 1). Our systematic searches yielded a total of 150 records. After excluding repetitive studies, a total of 114 articles were retrieved. Then, read the titles and abstracts of each of the retrieved articles to excluded unrelated animal models, unrelated interventions and unrelated research types. Finally, 10 animal studies were selected for the meta-analysis.

### 3.2 Characteristics of the included studies

The detailed characteristics of the included studies were shown in Table 1. Female CBA/J strain mice mated with the DBA/2 male mice were used as the abortion-prone mouse model in all 10 studies. For the source of MSCs, all of included studies used allogeneic MSCs for administration, in which four studies used bone mesenchymal stem cells (bMSCs) harvested from the femur or/and tibial bone marrow, six studies used adipose-derived stem cells (AD-MSCs) from nonpregnant female CBA/J mice. As for the MSC injection method, eight animal experiments used intraperitoneal injection, two experiments used caudal vein injection and two experiments used uterine horns for injection. Details of the basic characteristics of the included studies were shown in Table 1.

### 3.3 Risk of bias assessment

The major limitation was that most of those did not report allocation concealment, random sequence generation, and blinding of participants and personnel. Although all studies reported

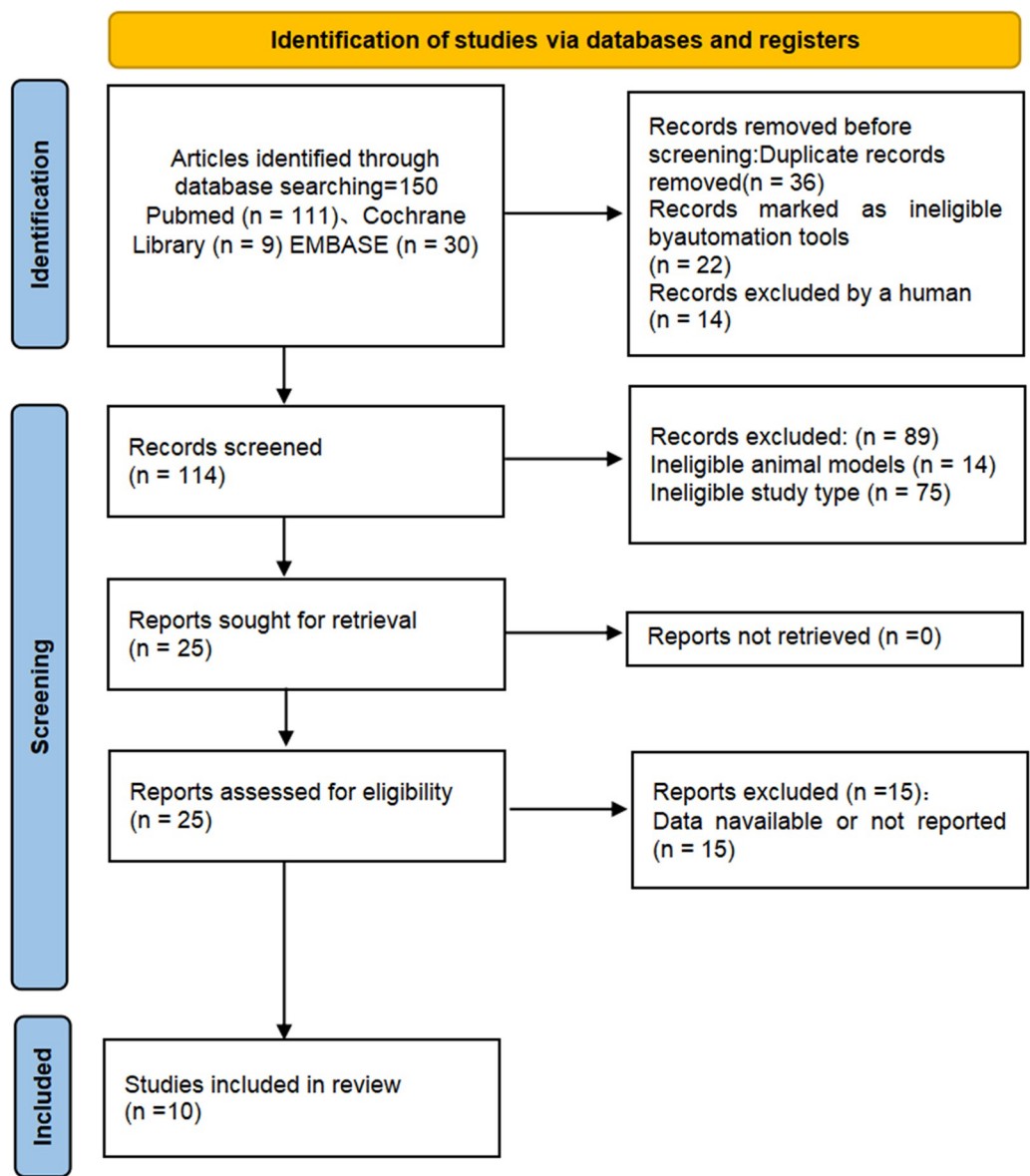

**Fig 1. Preferred Reporting Items for Systematic Reviews and Meta-analysis (PRISMA) flow diagram of systematic search.**

that animals were randomized to experimental groups, no studies provided details regarding method of randomization which is essential to assess adequate random sequence generation. Six studies displayed a low risk for reporting baseline characteristics. However, the risk of bias was high across all studies for the random housing and blinding of personnel which meant there maybe exhibit selection bias and performance bias. Incomplete outcome data (i.e. attrition bias) was assessed as whether the N-value was consistent between the method and result sections, nevertheless, all studies were classified as high risk. Lastly, other potential sources of bias considered included source of funding, sample size calculation and conflict of interests, ten studies were assessed as low risk (Table 2).

**Table 1. Characteristics of the included studies.**

| Study | Year | Country | Animal model | | Number of mice | | MSC source/ dosage | Administration | Terminology consistency | Positive surface markers | Negative surface markers |
|---|---|---|---|---|---|---|---|---|---|---|---|
| | | | T | C | T | C | | | | | |
| Meng [20] | 2016a | China | CBA/J×DBA/2 | | 10 | 10 | bMSCs, allogeneic, $1×10^8$/ml | Caudal vein | Yes | embryo absorption rate, IL4, IL10, IFN-γ, TNF-α | Not described |
| Meng [20] | 2016b | China | CBA/J×DBA/2 | | 10 | 10 | bMSCs, allogeneic, $1×10^8$/ml | Uterine horns | Yes | embryo absorption rate, IL4, IL10, IFN-γ, TNF-α | Not described |
| Eskandarian [21] | 2019 | Iran | CBA/J×DBA/2 | | 5 | 5 | AD-MSCs, allogeneic, $1×10^6$/ml | Intraperitoneal | Yes | embryo absorption rate | Not described |
| Farrokhi [22] | 2018 | Iran | CBA/J×DBA/2 | | 7 | 7 | AD-MSCs, allogeneic, $1×10^6$/ml | Intraperitoneal | Yes | embryo absorption rate, IL-4, IFN-γ | Not described |
| Farrokhi [11] | 2021a | Iran | CBA/J×DBA/2 | | 10 | 10 | AD-MSCs, allogeneic, $1×10^6$/ml | Intraperitoneal | Yes | embryo absorption rate, | Not described |
| Farrokhi [11] | 2021b | Iran | CBA/J×DBA/2 | | 10 | 10 | AD-MSCs, allogeneic, $1×10^6$/ml | Intraperitoneal | Yes | embryo absorption rate, IL4, IL10, IFN-γ | Not described |
| Rezaei [23] | 2020 | Iran | CBA/J×DBA/2 | | 5 | 5 | AD-MSCs, allogeneic, $1×10^6$/ml | Intraperitoneal | Yes | embryo absorption rate, IL4, IL10, IFN-γ | Not described |
| Xiang [24] | 2020 | China | CBA/J×DBA/2 | | 5 | 5 | bMSCs, allogeneic | Uterine horns | Yes | embryo absorption rate, IL4, IL10, IFN-γ, TNF-α | Not described |
| Kahmini [25] | 2019 | Iran | CBA/J×DBA/2 | | 5 | 5 | AD-MSCs, allogeneic, $1×10^6$/ml | Intraperitoneal | Yes | embryo absorption rate | Not described |
| Sadighi [26] | 2018 | Iran | CBA/J×DBA/2 | | 5 | 5 | AD-MSCs, allogeneic, $1×10^6$/ml | Intraperitoneal | Yes | embryo absorption rate | Not described |
| Shahgaldi [27] | 2022 | Iran | CBA/J×DBA/2 | | 7 | 7 | AD-MSCs, allogeneic, $1×10^6$/ml | Intraperitoneal | Yes | embryo absorption rate | Not described |
| Zhang [28] | 2021 | China | CBA/J×DBA/2 | | 6 | 6 | bMSCs, allogeneic, $2×10^6$/ml | Caudal vein | Yes | embryo absorption rate, IL10 | Not described |

**Abbreviations:** bMSCs: Bone marrow stromal cell; AD-MSCs: Adipose-derived stem cells; IL4: Interleukin-4; IL10: Interleukin-10; IFN-γ: Interferon-γ; TNF-α: Tumor necrosis factor-α.

## 3.4 Primary outcome: Effect of MSC treatment on the embryo absorption rate

A total of nine studies reported the embryo absorption rate, as shown in Fig 2. The heterogeneity among the data was low, so the fixed effect model was used. The results showed that MSC treatment significantly reduced the embryo absorption rate (OR 0.23, 95%CI: 0.17, 0.32, $P<0.00001$).

## 3.5 Secondary outcomes

**3.5.1 Effect of MSC treatment on IL4 in decidual and spleen tissues.** It is believed that IL-4 produced at the materno-fetal interface, which inhibits Th1 type cytokines and stimulates

 

**Table 2. Individual risk of bias and quality assessment using SYRCLE (Systematic Review Centre for Laboratory Animal Experimentation).**

| Study | Year | Baseline characteristics | Allocation concealment | Random housing | Performance blinding | Blinding of outcome assessment | Random outcome assessment | Incomplete outcome data | Selective outcome reporting | Other bias |
|---|---|---|---|---|---|---|---|---|---|---|
| Meng | 2016 | Low risk of bias | Low risk of bias | High risk of bias | Unclear | Unclear | Low risk of bias | High risk of bias | Low risk of bias | Low risk of bias |
| Rezaei | 2020 | Low risk of bias | Low risk of bias | High risk of bias | Unclear | Unclear | Low risk of bias | High risk of bias | Low risk of bias | Low risk of bias |
| Eskandarian | 2019 | Unclear | Low risk of bias | High risk of bias | Low risk of bias | Unclear | Unclear | High risk of bias | High risk of bias | Low risk of bias |
| Farrokhi | 2018 | Unclear | Low risk of bias | High risk of bias | Low risk of bias | Unclear | Unclear | High risk of bias | Low risk of bias | Low risk of bias |
| Farrokhi | 2021 | Unclear | Low risk of bias | High risk of bias | Unclear | Unclear | Unclear | High risk of bias | Low risk of bias | Low risk of bias |
| Xiang | 2020 | Low risk of bias | Low risk of bias | High risk of bias | Unclear | Unclear | Low risk of bias | High risk of bias | Low risk of bias | Low risk of bias |
| Kahmini | 2019 | Low risk of bias | Low risk of bias | High risk of bias | Unclear | Unclear | Low risk of bias | High risk of bias | Low risk of bias | Low risk of bias |
| Sadighi | 2018 | Low risk of bias | Low risk of bias | High risk of bias | Unclear | Unclear | Low risk of bias | High risk of bias | Low risk of bias | Low risk of bias |
| Shahgaldi | 2022 | Unclear | Low risk of bias | Low risk of bias | Unclear | Unclear | Unclear | Low risk of bias | Low risk of bias | Low risk of bias |
| Zhang | 2021 | Low risk of bias | Low risk of bias | High risk of bias | Unclear | Unclear | Low risk of bias | High risk of bias | Low risk of bias | Low risk of bias |

B cell production of antibodies, is helpful for maintaining pregnancy [29]. Thus, we explore the effect of MSC treatment on IL4 in decidual and spleen tissues. A totatl of eight studies reported IL4 level changes after MSC treatment, in which 5 articles observed IL4 in decidual tissues, and the rest three article detected IL4 in spleen tissues. Random effect model was utilized on account of the high heterogeneity among studies. And the results displayed that the expression profile of IL4 in decidual tissues was higher in the experimental group than in the

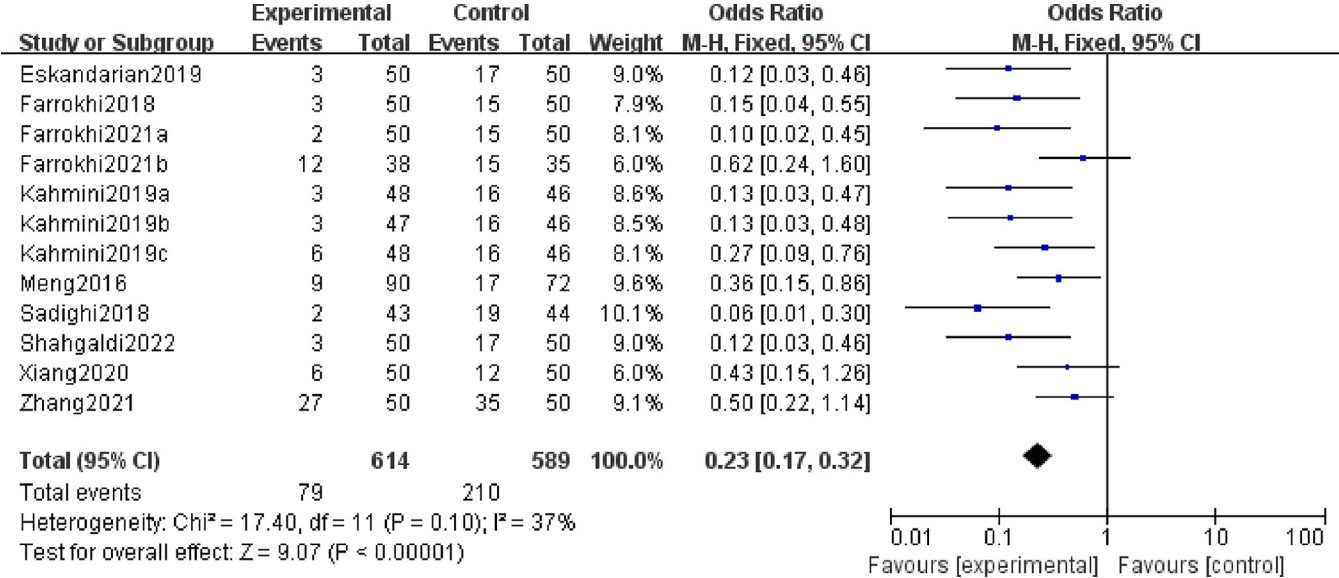

**Fig 2. The forest plot of the embryo resorption rate between the experimental group and control group.**

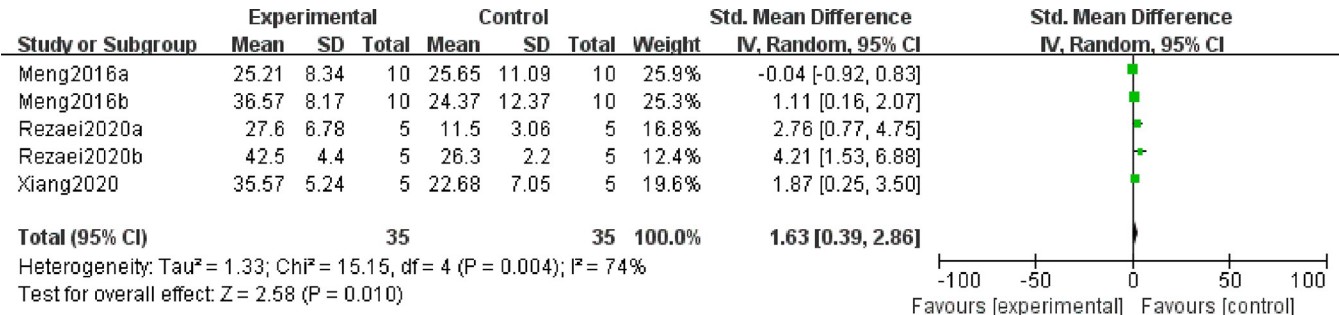

**Fig 3. Forest plot on the expression of IL4 in decidual tissue between the experimental group and control group.**

control group (SMD = 1.63, 95%CI: 0.39, 2.86, *P* = 0.01, Fig 3). However, no significant differences were found in the levels of IL4 from the spleen between the experimental and control groups (SMD = 0.09, 95%CI: -0.46, 0.65, *P* = 0.74, S1A Fig). Because of the high heterogeneity among these five articles that reported IL4 levels derived from decidual tissues, subgroup analysis were performed according to the administration route. The result revealed that both cornua uteri and intraperitoneal could significantly improve the expression of IL4 from decidual tissue with low heterogeneity (SMD = 1.31, 95%CI: 0.48, 2.14, *P* = 0.002; SMD = 3.28, 95%CI: 1.68,4.87, *P*<0.0001, Fig 4), while one study reported that there was no difference in caudal vein injection and no MSC treatment (SMD = -0.04, 95% CI: -0.92,0.83, *P* = 0.92, Fig 3). Thus, it can be conferred that MSCs may be an essential therapy for IL-4 reduction in the decidua of patients with URSA in the future.

**3.5.2 Effect of MSC treatment on IL10 in decidual and splenic tissues.** IL-10 modulates the balance of pro-inflammatory and anti-inflammatory signals on the endometrium to

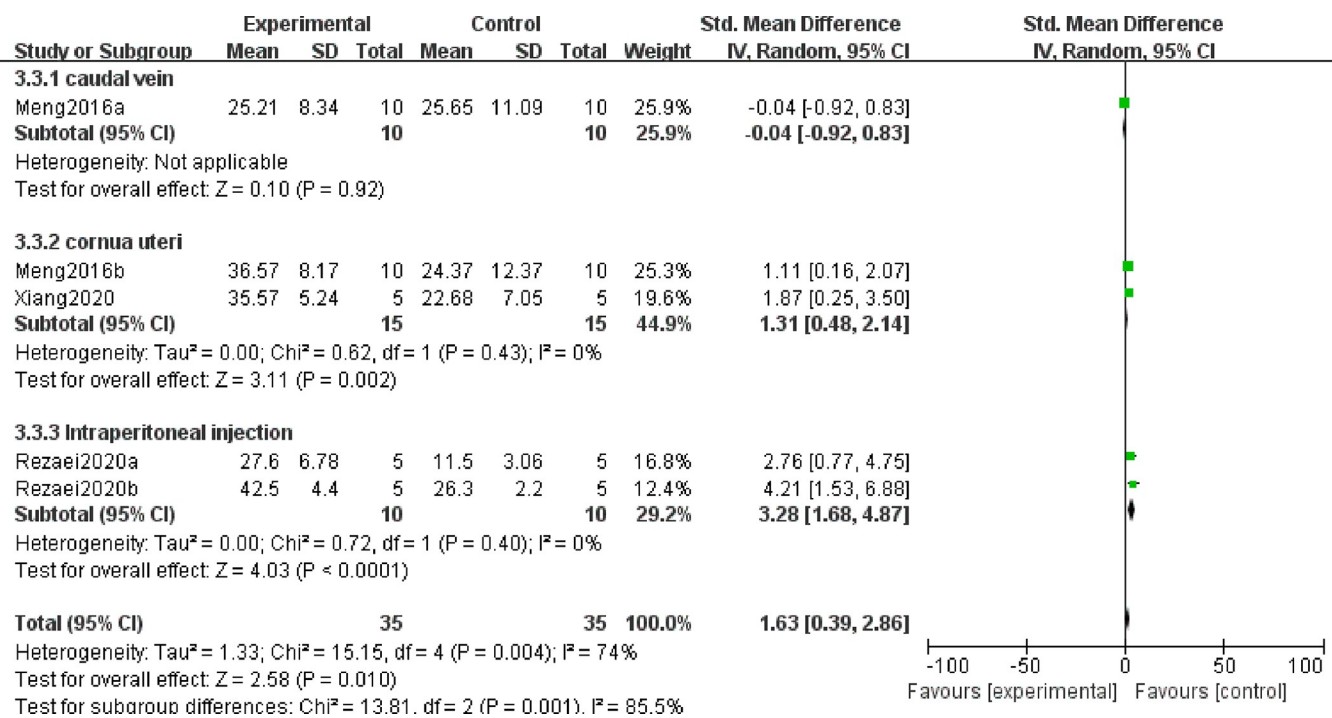

**Fig 4. Forest plots for the effects in different cite of injection cells on IL4 from decidual tissue.**

**Fig 5. Forest plot on the expression of IL10 from decidual tissue between the experimental group and control group.**

protect tissue from damage caused by inflammatory responses, which is vital in maintaining normal pregnancy [30]. Regarding IL10 levels, six articles observed IL10 in decidual tissues, and three articles reported IL10 in spleen tissues. A random effect model was chosen due to the high heterogeneity among studies, and the results suggested that MSC treatment might increase IL10 expression in the decidual tissues (SMD = 1.60, 95%CI: 0.58, 2.61, $P$ = 0.002, Fig 5), while no significant differences were found in the levels of spleen IL10 between the experimental and control groups (SMD = 0.01, 95%CI: -0.54, 0.57, $P$ = 0.96, S1B Fig). To investigate the heterogeneous origin of IL10 levels in decidual tissues, subgroup analysis were performed. The result hinted that different injection routes were a source of heterogeneity. And a significant difference between each subgroup was observed with high heterogeneity ($P$ = 0.009, $I^2$ = 78.9%, Fig 6A). Results from subgroup revealed that both cornua uteri and intraperitoneal injection could significantly improve IL10 in decidual tissues (SMD = 2.14, 95%CI: 0.45, 3.83, $P$ = 0.01; SMD = 2.81, 95%CI: 1.37, 4.25, $P$ = 0.0001, Fig 6A). Besides, we also conducted subgroup analysis according to the MSC tissue source, and the results indicated that both bMSCs and AD-MSCs had a positive adjustment for the expression level of IL10 (SMD = 1.13, 95%CI: 0.08, 2.17, $P$ = 0.03; SMD = 2.81, 95%CI: 1.37, 4.25, $P$ = 0.0001, Fig 6B). These showed that MSCs were a promising way of increasing IL-10 levels in the decidua of URSA patients.

**3.5.3 Effect of MSC treatment on IFN-γ in decidual and spleen tissues.** IFN-γ is an essential factor leading to miscarriage, as it can cause human trophoblast cells to undergo apoptosis and suppress their proliferation [31]. In this study, we observed the effect of MSC treatment on IFN-γ in decidual and spleen tissues. A total of 5 studies reported the IFN-γ level in the decidual tissues, and three studies reported IFN-γ in spleen tissues. Because of the high heterogeneity among studies, random effect model was used. The results showed that the level of decidual IFN-γ in the experimental group was significantly lower than that in the control group (SMD = -1.66, 95%CI: -2.79, -0.52, $P$ = 0.004, Fig 7). Nevertheless, no significant differences were found in spleen IFN-γ between groups (SMD = -0.19, 95%CI: -0.75, 0.36, $P$ = 0.5, S1C Fig). Then, subgroup analysis was performed according to injection method which might be the source of heterogeneity between groups, and the result showed that caudal vein injection, cornua uteri injection and intraperitoneal injection were all capable of decreasing the expression of IFN-γ in decidual tissues (SMD = -0.12, 95%CI: -1.00, 0.76, $P$ = 0.76; SMD = -1.63, 95%CI: -2.50, -0.77, $P$ = 0.0002; SMD = -2.92, 95%CI: -4.57, -1.26, $P$ = 0.0005, Fig 8A). Besides, we also conducted sub-analysis according to stem-cell origin, and the results showed that AD-MSCs had a tendency to reduce the expression of IFN-γ in the decidual tissue of URSA (SMD = -2.92, 95%CI: -4.57, -1.26, $P$ = 0.0005, Fig 8B). While, the efficacy of bMSCs on IFN-γ was similar to the control group (SMD = -1.06, 95%CI: -2.18, 0.06, $P$ = 0.06, Fig 8B).

**(A)**

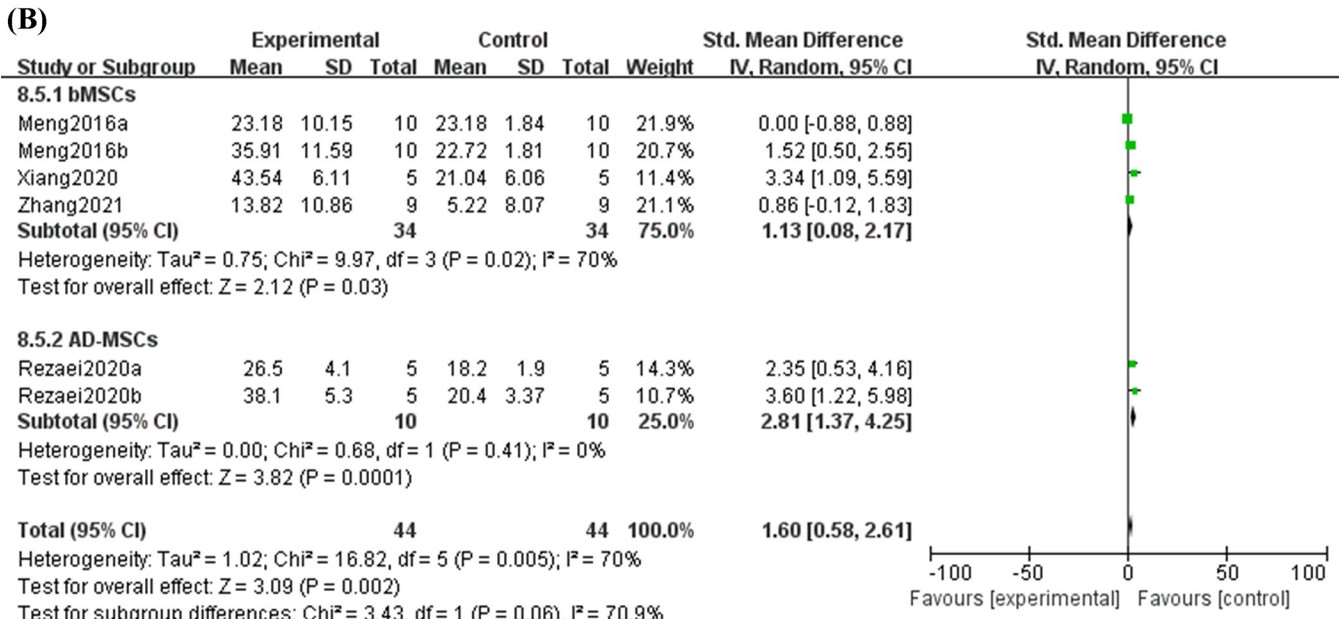

**(B)**

**Fig 6. Subgroup analysis of MSC treatment effects on the expression of IL10 from decidual tissue.** (A) Injection method; (B) Sources of stem cells.

These suggested that MSCs could be a valuable tactic for lowering IFN-γ in the decidua of females with URSA.

**3.5.4 Effect of MSC treatment on TNF-α in decidual and spleen tissues.** TNF-α may directly promote tissue damage in pregnancy and activate maternal monocytes bound to

| Study or Subgroup | Experimental | | | Control | | | Weight | Std. Mean Difference IV, Random, 95% CI | Std. Mean Difference IV, Random, 95% CI |
|---|---|---|---|---|---|---|---|---|---|
| | Mean | SD | Total | Mean | SD | Total | | | |
| Meng2016a | 26.13 | 11.32 | 10 | 27.23 | 4.53 | 10 | 26.4% | -0.12 [-1.00, 0.76] | |
| Meng2016b | 16.82 | 4.53 | 10 | 26.15 | 6.72 | 10 | 24.9% | -1.56 [-2.59, -0.53] | |
| Rezaei2020a | 26.7 | 5.7 | 5 | 41.2 | 5.71 | 5 | 17.6% | -2.30 [-4.09, -0.50] | |
| Rezaei2020b | 35.45 | 4.06 | 5 | 51.7 | 3.1 | 5 | 11.8% | -4.06 [-6.67, -1.46] | |
| Xiang2020 | 18.38 | 5.59 | 5 | 28.38 | 4.27 | 5 | 19.2% | -1.82 [-3.42, -0.21] | |
| **Total (95% CI)** | | | 35 | | | 35 | 100.0% | -1.66 [-2.79, -0.52] | |

Heterogeneity: Tau² = 1.07; Chi² = 12.91, df = 4 (P = 0.01); I² = 69%
Test for overall effect: Z = 2.86 (P = 0.004)

Favours [experimental]  Favours [control]

**Fig 7. Forest plot on the expression of IFN-γ from decidual tissue between the experimental group and control group.**

receptors on the placenta, inducing apoptosis, which may cause URSA [32]. Here, we assessed the effect of MSC treatment on TNF-αin decidual and spleen tissues. A total of two studies reported the expression of TNF-α detected from decidual tissues. The fixed effect model was used for combined analysis accounting for low heterogeneity. The results verified that there was a significant difference in decidual TNF-α level between the MSC treatment group and control group (SMD = -1.98, 95%CI: -2.93, -1.04, $P < 0.0001$, Fig 9), while there was no difference in the spleen TNF-α level between groups (SMD = -0.18, 95%CI: -0.73, 0.38, $P = 0.53$, S1D Fig). Thus, It will be evident from this that MSC-based cell treatment may effectively lower TNF-α in the decidua of URSA patients.

### 3.6 Publication bias

The funnel plot for all the outcomes we measured were scattered and asymmetric (S2 Fig). Thus, Egger's test was used to evaluated for the publication bias of the embryo absorption rate (n = 10), which demonstrated publication bias ($P<0.001$, Fig 10). However, due to the presence of heterogeneity, the statistical assessment of publication bias is unreliable.

## 4 Discussion

This systematic review and meta-analysis illustrated that MSCs markedly decreased the embryo resorption rate in URSA mouse models. Additionally, MSC treatment positively improved the levels of IL4 and IL10 and inversely affected the expression of TNF-α and IFN-γ in decidual tissues. A further subgroup analysis was conducted to compare the impacts of disparate administration routes and MSC sources on MSC therapy. Notably, intraperitoneal injection and cornua uteri administration routes appeared to be particularly effective in enhancing the expression of IL4, IL-10 and reducing IFN-γ levels in decidual tissues. Furthermore, there was no significant difference observed in the modulation of IL4, IL10, IFN-γ between bMSCs and AD-MSCs. This comprehensive examination of MSC therapy's impact on URSA in animal models revealed beneficial effects, contributing to an expansive preclinical data pool regarding MSC's potential application in human URSA.

The accomplishment of a successful pregnancy is largely contingent upon an accurate immunologic dialogue at the maternal-fetal immune interface in the endometrium [33]. Conversely, URSA, as a severe reproductive disorder, is characterized by immune dysfunction related to the breakdown of fetal-maternal immune tolerance. Given the semi-allogeneic nature of the fetus, the decidual immune cell network takes charge of regulating the immune response at the fetomaternal interface [34] and shields the embryo from the mother's immunological reactions [35]. Numerous studies corroborated the finding of abnormal immune cell proportions, such as CD4+ T cells, Treg Tcells and natural killer (NK) cells, in the decidual

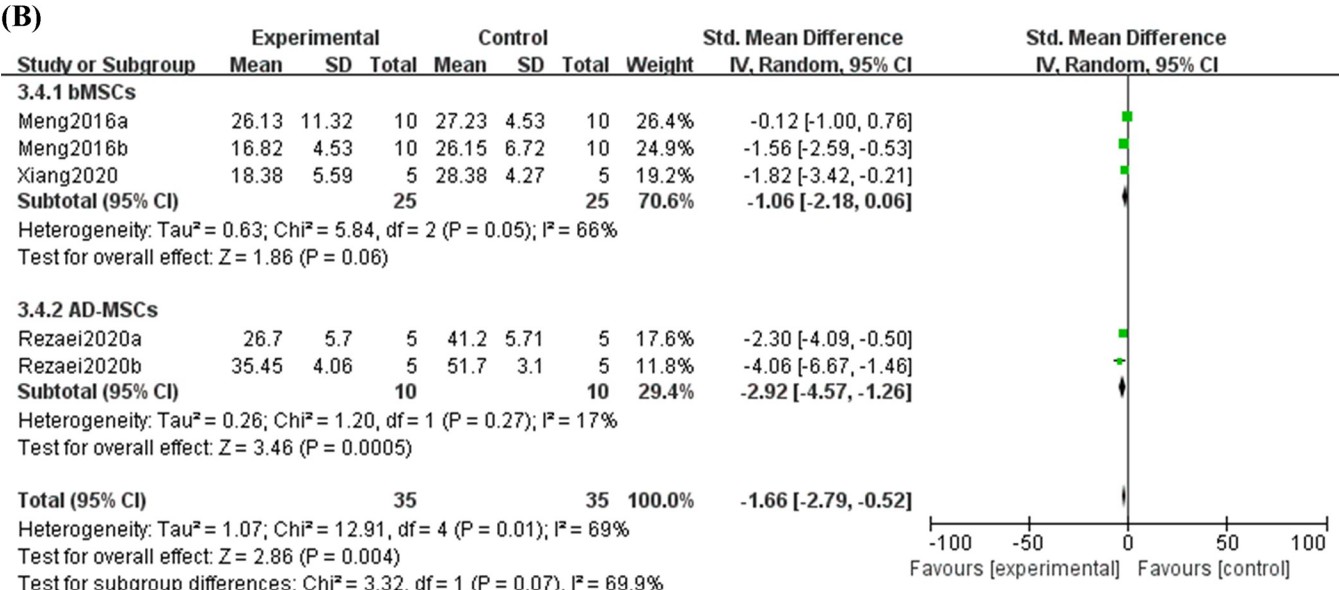

**Fig 8. Subgroup analysis of MSC treatment effects on the expression of IFN-γ from decidual tissue.** (A) Injection method; (B) Sources of stem cells.

tissues of pregnant women diagnosed with URSA [36–38]. For instance, it has been reported that the ratios of Th1, Th17, NK cells in URSA patients during the first trimester were elevated [39,40]. Conversely, the proportions of Th2, Tregs in the uterine decidua of URSA were lower than in normal pregnancies. This heightened inflammatory immune response during pregnancy, common in most women with URSA, paralleled the pregnancy loss model of the CBA/ J × DBA/2 mating model, which was closely aligned with altered immune mechanisms. Studies

| Study or Subgroup | Experimental Mean | SD | Total | Control Mean | SD | Total | Weight | Std. Mean Difference IV, Fixed, 95% CI | Std. Mean Difference IV, Fixed, 95% CI |
|---|---|---|---|---|---|---|---|---|---|
| Meng2016b | 37.99 | 9.05 | 10 | 58.75 | 13.56 | 10 | 79.6% | -1.72 [-2.79, -0.66] | |
| Xiang2020 | 47.35 | 4.6 | 5 | 62.61 | 4.6 | 5 | 20.4% | -3.00 [-5.09, -0.90] | |
| Total (95% CI) | | | 15 | | | 15 | 100.0% | -1.98 [-2.93, -1.04] | |

Heterogeneity: Chi² = 1.13, df = 1 (P = 0.29); I² = 11%
Test for overall effect: Z = 4.11 (P < 0.0001)

Favours [experimental]   Favours [control]

**Fig 9. Forest plot on the expression of TFN-α from decidual tissue between the experimental group and control group.**

have revealed that in DBA/2-mated CBA/J females, there was a loss of cellular contact between decidual cells and cells of the ectoplacental cone beginning around gestation day 7, accompanied by disorganized infiltrating immune cells at the maternal-fetal interface, such as Th1/Th2/Th17/Treg cell imbalance and defects in macrophage, T-cell, and NK cell function [41,42]. Therefore, this pattern was deemed most analogous to human URSA during the first trimester. Hence, in this meta-analysis, all included studies employed the CBA/J × DBA/2 mating model to examine the effects of MSC treatment on URSA.

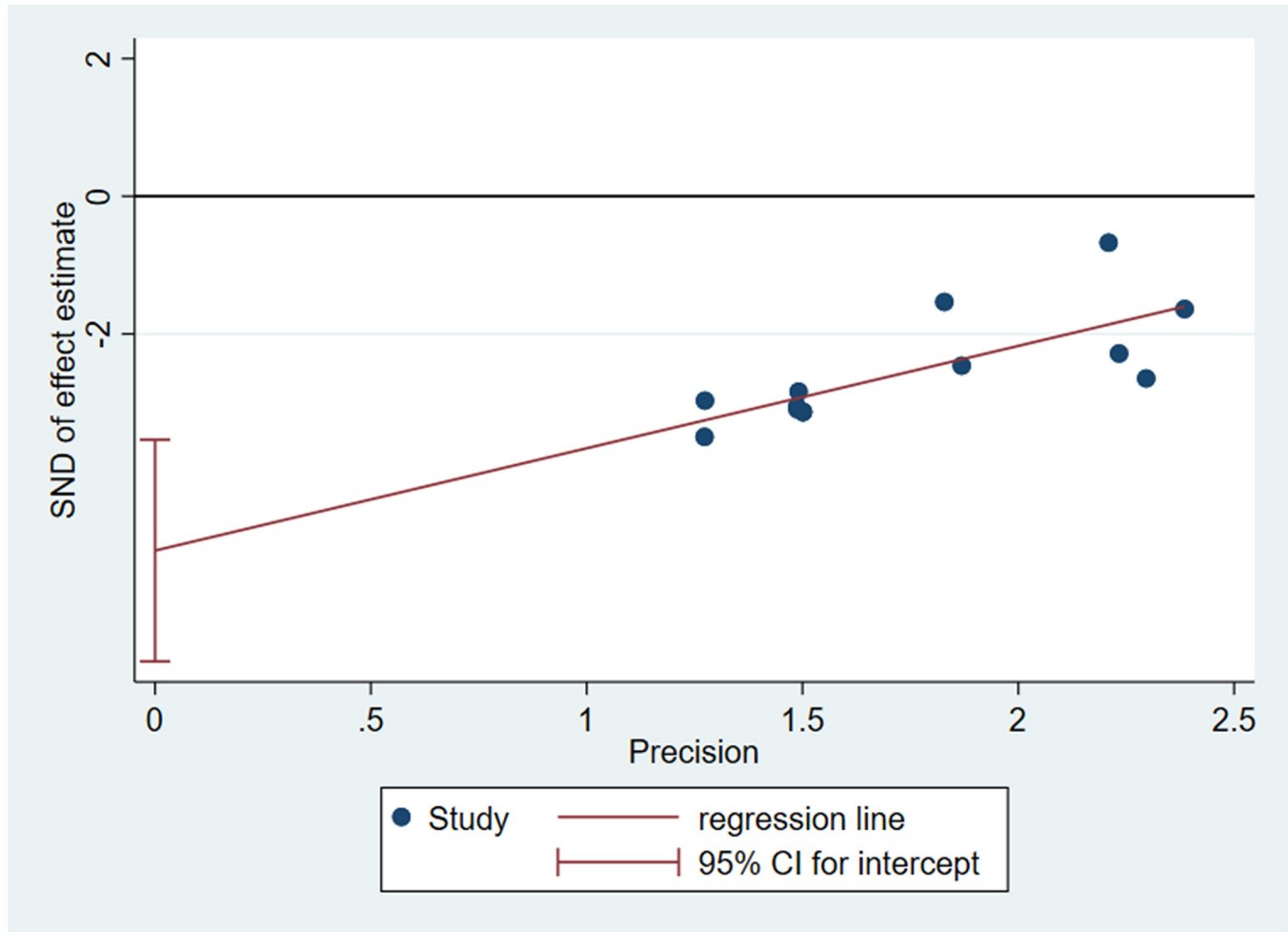

**Fig 10. The assessment of possible bias: Egger's publication bias plot for embryo absorption rate.**

Stem cells have been identified as potential regulators of cellular immunity across numerous autoimmune diseases [43]. Currently, the application of of MSC treatment in recurrent spontaneous abortion (RSA) remains nascent, compelling evidence suggests that MSCs orchestrate Th1/Th2 balance via contact and the excretion of immunomodulatory factors [44,45]. This regulation has been seen to inhibit proliferation and cytotoxicity of Natural Killer (NK) cells [46], thus establishing an immune balance at the maternal-fetal interface in RSA mouse models. The current meta-analysis revealed significant differential expressions of IL4, IL10, IFN-γ, and TNF-α in the decidua tissue of the URSA murine model post-MSC treatment (P< 0.05). Notably, IFN-γ, predominantly secreted by NK and Th1 cells within the uterus, facilitates decidual NK cell differentiation, placental formation, and decidua maintenance at the maternal-fetal interface—all critical to successful pregnancy [47]. However, in the endometrium and decidua of RSA patients, elevated IFN-γ interplays with excessive inflammatory responses, thereby triggering cell apoptosis and inducing embryonic toxicity [48]. Our study shows that MSCs can mitigate IFN-γ levels at the URSA maternal-fetal interface, potentially shielding cells from rampant apoptosis, thereby demonstrating an embryo-protective role. As for TNF-α, it is mainly present in decidua and is secreted by uterine NK cells, neutrophil granulocytes and T lymphocytes [49]. It has been confirmed that its moderate release at the maternal-fetal interface improves the development of tolerant uterine dendritic cells, thus fostering maternal embrace of the embryo [50]. Conversely, its excessive release can instigate trophoblast cell apoptosis, activate the coagulation system, culminating in the formation of thrombosis in the placental vessels and subsequent adverse pregnancy outcomes [51]. Our study revealed that MSCs could reduce TNF-α levels in URSA decidual tissues, which could be beneficial to trophoblasts and assist in recasting blood vessels at the maternal-fetal interface. IL10 and IL4 function as anti-inflammatory cytokines, with IL10 predominantly secreted by dendritic cells, macrophages, mast cells, and natural killer cells, and IL-4 primarily secreted by T cells, mast cells, and basophils. These cytokines are integral for stimulating B and T cell proliferation and reducing IL-12 production in dendritic cells [52]. Both IL4 and IL10 act as vital immunosuppressive factors in the decidual tissue during normal pregnancies, modulating communication between embryonic trophoblasts and maternal decidual cells and inhibiting the secretion of IFN-γ and TNF-α to avert acute immune rejection [53]. In URSA cases, however, a marked reduction in IL-4 and IL10 cytokine production in decidual tissue disrupts the immune environment at the maternal-fetal interface, provoking maternal immune rejection of the embryo and resultant pregnancy loss. This study demonstrates that MSCs could augment IL10 and IL4 levels at the maternal-fetal interface, thereby potentially promoting immune tolerance. Consequently, the results corroborate the advantageous immunomodulatory effects of MSC treatment in the URSA murine model, indicating that MSC-based cell therapy holds promise for URSA.

In light of the lack of a universally agreed optimal cell type or treatment regimen for RSA, our subgroup analysis scrutinized the impact of MSC source and delivery route choices on RSA pregnancy outcomes. Regarding MSC sources, the studies incorporated in this analysis utilized bMSC or AD-MSC. Both bMSC and AD-MSC, often employed stem cells, could migrate to injury or inflammation sites, differentiate into local cells, and secrete chemokines, cytokines, and growth factors that facilitate tissue regeneration and exert anti-inflammatory effects [54]. Besides, both cell types also exhibited robust differentiation capabilities. The current study deduced that bMSCs were the most commonly used source for generating spheroids, followed by AD-MSCs. It has been established that bMSCs could engender uterine endothelial progenitors, thereby aiding in maintaining the plasticity and integrity of the maternal-fetal interface during pregnancy [55,56]. Conversely, AD-MSCs also presented several distinct benefits. They displayed superior proliferation capabilities, growth factor secretion

capacities, and had an extended lifespan compared to bMSCs [57]. Additionally, compared to bMSC, AD-MSC could be collected through liposuction, which was less invasive to the donor. Additionally, AD-MSC had a 500-fold higher concentration than bMSC, indicating that it was easier to obtain AD-MSC than bMSC [58,59]. Drawing from the aforementioned literature, it could be concluded that both stem cell types had significant potential for clinical application. Our meta-analysis demonstrated that both AD-MSCs and bMSCs were effective in mitigating IFN-γ expression and enhancing IL10 levels in the decidual tissue of URSA mice, thereby serving a multi-faceted immunomodulatory role in URSA.

Furthermore, concerning the delivery route, the studies encompassed within this meta-analysis utilized three routes, namely tail vein injection, intraperitoneal injection, and uterine horn injection. The analysis confirmed that both intraperitoneal and uterine injections could significantly modulate IL4, IL10, and IFN-γ levels in a manner beneficial for pregnancy, whereas no notable difference was observed between intravenous administration and non-MSC treatment. Currently, owing to the convenience and minimal trauma, intravenous and tail vein injections are the most frequently used routes of injection in systemic administration. Animal experiments have illustrated that labelled MSCs could be located within the peritoneum, abdomen, and uterine adipose tissue of pregnant mice following intraperitoneal MSC injection [60]. However, MSCs injected through vein tend to be filtered by other organs within the cellular pathway [61–63], particularly the lungs. Cheng et al. determined that 62% of MSCs concentrated in the lungs one hour post intravenous transplantation, with negligible recovery in other tissues [64–66]. As for uterine horn injection, this localized injection route permits direct MSC recruitment into the uterine cavity, potentially overcoming the lower homing cell incidence associated with systemic administration. A preceding study [67], using a mouse model, examining the impact of MSCs on ovarian hypofunction, suggested that local injection was more suitable than intravenous injection. These findings align with our results, suggesting that in the clinical management of MSCs in URSA, strategies to amplify local MSC concentration, such as intrauterine perfusion, should be employed.

Inevitably, our meta-analysis bears limitations meriting discussion. First, current models of RSA only simulate aspects of human URSA, thus constraining their translational capacity. Additionally, as with numerous studies, this meta-analysis was conducted amid significant heterogeneity. Therefore, there is a pressing need for further high-caliber animal studies employing double-blinding and randomization techniques to minimize bias. Despite these constraints, our meta-analysis consistently demonstrated the beneficial effects of MSC therapy, with 95% CIs overlapping and supporting the findings. Moreover, factors such as cell dosage, number of administrations, and delivery route are presumed to impact MSC therapy. However, these aspects remain under-investigated in URSA. Consequently, our study primarily concentrates on two types of stem cells and three transplantation routes due to the limitations of existing research, potentially overlooking the therapeutic potential of stem cells from alternative sources and delivery routes.

In spite of the limitations of these animal studies, they provide insights into possibilities of MSCs for human URSA. To the best of our knowledge, this is the first study in the field to comprehensively assess MSCs in the treatment of URSA from the perspectives of embryo uptake rate and cytokine synthesis. These findings, based on preliminary evidence from animal studies, may optimize therapeutic strategies for URSA clinical trials and guide the design of future animal experiments.

## Conclusion

Overall, this systematic review and meta-analysis provides insurmountable evidence for the efficacy of MSCs in the experimental animal models of URSA. MSC therapy improved

immunologic derangement inherent in URSA pathogenesis and progression for which MSC tissue source and administration route further modified this effect. These quantitatively summarized preclinical findings are conducive to guiding therapy selection and successful translation of findings to clinical trials. In future, pre-clinical and clinical research of URSA is also required to ascertain the optimal MSC delivery pathway for maximum therapy success and create valuable biomarkers.

## Supporting information

**S1 Fig. Forest plot of IL4, IL10, IFN-γ and TFN-α from spleen tissue between experimental group and control group.**
(TIF)

**S2 Fig. The funnel plot for the embryo absorption rate, the expression of IL4, IL10, IFN-γ and TFN-α from decidual tissue between experimental group and control group.**
(TIF)

**S1 Table. PRISMA checklist.**
(PDF)

**S2 Table. Search strategy.**
(PDF)

## Acknowledgments

We thank all staffs involved in this study.

## Author Contributions

**Data curation:** Yijie Hu, Wenjun Xiao, Yiming Ma, Dan Shen.

**Methodology:** Xiaoxuan Zhao, Yi Shen, Suxia Wang.

**Writing – original draft:** Xiaoxuan Zhao, Yijie Hu, Wenjun Xiao.

**Writing – review & editing:** Xiaoxuan Zhao, Yuepeng Jiang, Jing Ma.

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
