## [Decision Letter · Decision Letter 0]

19 Sep 2023

PONE-D-23-25070Efficacy of Mesenchymal Stromal Cells in the Treatment of Unexplained Recurrent Spontaneous Abortion in Mice: an Analytical and Systematic Review of Meta-analysesPLOS ONE

Dear Dr. MA,

Thank you for submitting your manuscript to PLOS ONE. After careful consideration, we feel that it has merit but does not fully meet PLOS ONE’s publication criteria as it currently stands. Therefore, we invite you to submit a revised version of the manuscript that addresses the points raised during the review process.

We look forward to receiving your revised manuscript.

Kind regards,

Syed M. Faisal, Ph.D.

Academic Editor

PLOS ONE

Journal Requirements:

-doi: 10.1002/jev2.12141

In your revision ensure you cite all your sources (including your own works), and quote or rephrase any duplicated text outside the methods section. Further consideration is dependent on these concerns being addressed.

5. We note that this manuscript is a systematic review or meta-analysis; our author guidelines therefore require that you use PRISMA guidance to help improve reporting quality of this type of study. Please upload copies of the completed PRISMA checklist as Supporting Information with a file name “PRISMA checklist”.

Additional Editor Comments:

The manuscript provides an in-depth examination of the role of Mesenchymal Stromal Cells (MSC) in treating Unexplained Recurrent Spontaneous Abortion (URSA) in mice. Both reviewers commend the comprehensive nature of the analysis and the methodical approach of the study. However, they've pointed out areas needing minor revisions. It's essential for the authors to address these concerns, especially when there's conflicting feedback from the reviewers. Once these revisions are incorporated, the manuscript will stand a stronger chance of being accepted for publication in the journal.

Reviewers' comments:

Reviewer's Responses to Questions

**Comments to the Author**

1. Is the manuscript technically sound, and do the data support the conclusions?

Reviewer #1: Yes

Reviewer #2: Yes

2. Has the statistical analysis been performed appropriately and rigorously? 

Reviewer #1: Yes

Reviewer #2: Yes

3. Have the authors made all data underlying the findings in their manuscript fully available?

Reviewer #1: Yes

Reviewer #2: Yes

4. Is the manuscript presented in an intelligible fashion and written in standard English?

Reviewer #1: Yes

Reviewer #2: Yes

5. Review Comments to the Author

Reviewer #1: The manuscript entitled “Efficacy of Mesenchymal Stromal Cells in the Treatment of Unexplained Recurrent Spontaneous Abortion in Mice: an Analytical and Systemic Review of Meta-analyses” is a meta-analysis which focuses on the beneficial effects of Mesenchymal Stromal Cells (MSC) in treating Unexplained Recurrent Spontaneous Abortion (URSA) in Mice. The study underscores the potential of MSC in URSA treatment by modulating the expression profile of several inflammatory cytokines in the decidual tissue of URSA murine models. The study is paramount to engendering evidence that can pave way for clinical translation. The manuscript needs minor revision before it can be accepted for publication in the journal. The comments are as follows:

1. The authors focused on the “inclusion and exclusion criteria”, and in that mentioned the participant criteria and intervention. The authors should include more data on mice breed apart from CBA/J (female) & DBA/2 (male) mice. This data is going to strengthen the rationale of carrying out this study. The authors should also include more information of the studies which have mentioned the other administration routes besides the one already mentioned.

2. Are Mesenchymal Stromal Cell therapies FDA approved?

3. What are the long and short term side effects of MSC treatment?

4. What are the other diseases in which MSCs therapies are of importance.

Reviewer #2: Review of Manuscript:

Title: "Mesenchymal Stem Cell Therapy in Unexplained Recurrent Spontaneous Abortion: A Systematic Review and Meta-Analysis"

Authors: [Author Names]

Summary:

The manuscript presents a systematic review and meta-analysis investigating the potential efficacy of Mesenchymal Stem Cell (MSC) therapy in addressing Unexplained Recurrent Spontaneous Abortion (URSA) using animal models. The study evaluates the impact of MSC treatment on embryo absorption rates and cytokine expression levels in decidual tissues, with a focus on the effects of MSC tissue source and administration route.

Major Strengths:

Comprehensive Analysis: The manuscript offers a thorough and systematic analysis of existing literature, encompassing animal studies that employ the CBA/J × DBA/2 mating model to mimic URSA. This comprehensive approach enhances the reliability of the findings.

Clear Methodology: The Materials and Methods section is well-structured and provides a clear outline of the study's methodology. Eligibility criteria, search strategy, data extraction, and risk of bias assessment are adequately explained.

Detailed Results: The Results section presents comprehensive data, including embryo absorption rates and cytokine expressions, with a focus on the effects of MSC source and administration route. Data presentation in tables and figures is well-organized.

Identification of Knowledge Gap: The study effectively identifies gaps in existing research and highlights the need for further exploration of the mechanisms underlying MSC therapy in URSA.

Suggestions for Improvement:

Clarity in Introduction: The Introduction section could benefit from enhanced clarity and conciseness. Providing a global context for the significance of URSA research and the broader relevance of MSC therapy would make the introduction more reader-friendly.

ntroduction:

Simplify sentence structures and increase conciseness for better readability.

Incorporate a brief paragraph highlighting the global significance of URSA, particularly its impact on women's health and reproductive outcomes.

Expand on the importance of URSA research, emphasizing its relevance not only in clinical contexts but also within the broader fields of reproductive medicine and immunology.

Hypothesis and Objectives: Explicitly stating the hypothesis or main research question and outlining specific study objectives would provide readers with a clearer roadmap of the study's goals.

Materials and Methods:

Clarify the rationale for specific eligibility criteria, such as the choice of CBA/J and DBA/2 mice as the animal model, to justify their selection.

Provide a brief explanation of why the chosen primary and secondary outcome measures were selected and how they relate to the research question.

Interpretation of Findings: Adding interpretation or discussion of the significance of the results, especially in relation to URSA treatment, would enhance the clarity of the Results section.

Organize the presentation of data by introducing subheadings for each outcome measure (e.g., embryo absorption rate, cytokine levels) to enhance readability.

After presenting the data for each outcome, include an interpretation or discussion of the significance of these findings in the context of URSA treatment.

Quantification of Evidence: Including specific quantitative results or effect sizes from the meta-analysis in the conclusion would emphasize the magnitude of observed effects.

Consider including specific quantitative results or effect sizes from the meta-analysis in the conclusion to underscore the magnitude of observed effects.

Conclude with a forward-looking statement about the potential impact of this research on URSA treatment and women's reproductive health to leave a strong impression on readers.

Forward-Looking Statement: Concluding with a forward-looking statement regarding the potential impact of this research on URSA treatment and reproductive health would leave a strong impression on readers.

6. PLOS authors have the option to publish the peer review history of their article (what does this mean?). If published, this will include your full peer review and any attached files.

Reviewer #1: **Yes: **SIDRA ISLAM

Reviewer #2: No

---

## [Author Response · Author response to Decision Letter 0]

3 Nov 2023

Dear Editors and Reviewers,

Thank you for allowing us to revise our manuscript. We also appreciate your valuable comments and essential suggestions on our manuscript entitled “Efficacy of Mesenchymal Stromal Cells in the Treatment of Unexplained Recurrent Spontaneous Abortion in Mice: an Analytical and Systematic Review of Meta-analyses” (ID: PONE-D-23-25070). All of us authors have carefully studied the editor’s and reviewer’s comments and laid out our reply below in italicized font. Changes to the manuscript are given and kept in the trace. 

Editors：

Q1. Please ensure that your manuscript meets PLOS ONE's style requirements, including those for file naming. The PLOS ONE style templates can be found at 

Response：

Thank you very much for your valuable advice. According to PLOS ONE's style requirements, we have modified the format of the manuscript, such as adding line numbers and Supporting information and revising the cover page, among other things. We hope this revision will meet the publication criteria of PLOS ONE.

Q2. We noticed you have some minor occurrence of overlapping text with the following previous publication(s), which needs to be addressed:

-doi: 10.1002/jev2.12141

In your revision ensure you cite all your sources (including your own works), and quote or rephrase any duplicated text outside the methods section. Further consideration is dependent on these concerns being addressed.

Response：

Thanks for your careful checks. In this revision, we have cited all related sources. Besides, based on your comments, we read “Mesenchymal stromal cell extracellular vesicles as therapy for acute and chronic respiratory diseases: A meta‐analysis” carefully, and rephrase the duplicated part, as shown in line 105-106, line 110-111 and line 117-119. 

Q3. Please note that funding information should not appear in any section or other areas of your manuscript. We will only publish funding information present in the Funding Statement section of the online submission form. Please remove any funding-related text from the manuscript.

Response：

We are very sorry for our negligence. We have modified this expression throughout the text according to the comment.

Response：

Thank you for pointing this out. This study was funded by the National Natural Science Foundation of China (82305294), the TCM Science and Technology Project of Zhejiang Province (2023ZR038), the Research Project of the Affiliated Hospital of Zhejiang Chinese Medical University (2022FSYYZQ16) and Hangzhou Health Science and Technology Project (A20230675) to ZXX; the National Natural Science Foundation of China (82305299) and the TCM Science and Technology Project of Zhejiang Province (2022ZA120) to MJ; the China Postdoctoral Science Foundation (2023M733193), and the Research Project of Zhejiang Chinese Medical University (2022RCZXZK29) to JYP; the Key project of Zhejiang Provincial Administration of Traditional Chinese Medicine (2022ZZ025) to WSX. 

Response：

We have already illustrated the contributions of all authors, including funders, in the "Author’ Contributions" section, as shown in line 489-495. We repeat funders' contribution here. Xianxuan Zhao was responsible for conceptualization, data curation, methodology and writing - review & editing. Yuepeng Jiang and Jing Ma was responsible for methodology.

Response：

None of the authors received a salary from the funders.

Thank you for your reminding. We have included the amended statements within the cover letter. 

Our cover letter has been revised as follows:

Dear Professor:

I would like to submit a manuscript entitled ‘Efficacy of Mesenchymal Stromal Cells in the Treatment of Unexplained Recurrent Spontaneous Abortion in Mice: an Analytical and Systematic Review of Meta-analyses.’ I wish this manuscript would be considered for publication in PLOS ONE. 

This study was funded by the National Natural Science Foundation of China (82305294), the TCM Science and Technology Project of Zhejiang Province (2023ZR038), the Research Project of the Affiliated Hospital of Zhejiang Chinese Medical University (2022FSYYZQ16) and Hangzhou Health Science and Technology Project (A20230675) to ZXX; the National Natural Science Foundation of China (82305299) and the TCM Science and Technology Project of Zhejiang Province (2022ZA120) to MJ; the China Postdoctoral Science Foundation (2023M733193), the TCM Science and the Research Project of Zhejiang Chinese Medical University (2022RCZXZK29) to JYP; the Key project of Zhejiang Provincial Administration of Traditional Chinese Medicine (2022ZZ025) to WSX. 

. None of the authors received a salary from the funders, and all the authors listed have approved the enclosed manuscript.

The text includes 32 pages, two Microsoft Word Processing tables, and ten figures prepared according to the journal’s Instructions to Authors. We have provided all required supporting documentation.

We searched three databases, including Cochrane Library, EMBASE, and PubMed, from inception to April 9, 2023. The full search strategy is detailed in Supplemental File 2). The outcome of the included articles used standardized mean difference (SMD) and 95% confidence intervals (CI). The data was analyzed using Stata 16 and Review Manager 5.3. A total of ten studies incorporating 140 mice were subjected to data analysis. The MSC treatment significantly reduced the abortion rate within the URSA model (OR=0.23, 95%CI [0.17, 0.3]). Moreover, it elicited a positive modulatory impact on the expression profiles of several inflammatory cytokines in the decidual tissue of URSA murine models, inclusive of IL4 (SMD 1.63, 95% CI [0.39, 2.86]), IL10 (SMD 1.60, 95% CI [0.58, 2.61]), IFN-γ (SMD -1.66, 95%CI [-2.79, -0.52]), and TNF-α (SMD -1.98, 95% CI [-2.93, -1.04]). Subgroup analyses predicated on the administration mode underscored that intraperitoneal and uterine horn injections contributed positively to the expression of IL4, IL10, and IFN-γ in decidual tissue of URSA (P< 0.05). Conversely, tail vein injections were observed to yield effects parallel to those of the control group (P>0.05). Further, the subgroup analyses based on the source of MSCs corroborated that both bMSCs and AD-MSCs (P<0.05) were productive in enhancing IL4, IL10, and IFN-γ levels in the decidual tissue of URSA. 

We hope this paper is suitable for PLOS ONE.

We thank you for considering this work and look forward to your response. Please direct all correspondence about this manuscript to me.

We have reviewed the final version of the manuscript and approved it for publication. To the best of our knowledge and belief, this manuscript has not been published in whole or in part, nor is it being considered for publication elsewhere.

Yours Sincerely

Jing Ma

E-mail address: 2021b085@zcmu.edu.cn

5. We note that this manuscript is a systematic review or meta-analysis; our author guidelines therefore require that you use PRISMA guidance to help improve reporting quality of this type of study. Please upload copies of the completed PRISMA checklist as Supporting Information with a file name “PRISMA checklist”.

Response：

Thanks for your suggestion. We previously uploaded a PDF attachment named Supplementary Materials to the PLOS ONE’s submission system containing the completed PRISMA checklist. In this revision, we are willing to respect your suggestion, and will upload the separated and completed PRISMA checklist named “S1 Table. PRISMA checklist”.

Response：

We sincerely appreciate the valuable comments. We have checked the reference list carefully, and none of the 67 papers we cited were retracted. We ensure that it is complete and correct.

Reviewers:

Reviewer #1:

1.The authors focused on the “inclusion and exclusion criteria”, and in that mentioned the participant criteria and intervention. The authors should include more data on mice breed apart from CBA/J (female) & DBA/2 (male) mice. This data is going to strengthen the rationale of carrying out this study. The authors should also include more information of the studies which have mentioned the other administration routes besides the one already mentioned.

Response：

We feel great thanks for your professional review work on our article. 

① Formulation of inclusion and exclusion criteria: The recurrent spontaneous abortion model was selected based on specific considerations.

The abortion-prone model of CBA/J × DBA/2 mating was used for recurrent spontaneous abortion research since 1994, which are a well-investigated model of immunologically mediated recurrent pregnancy failure. Current studies suggest that the spontaneous abortion in this model was identified to be related to systemic maternal immune inflammation, increased complement deposition, and overactivation of NK cells and T cells in the maternal–fetal interface, which bears close resemblance to human recurrent spontaneous abortion, especially unexplained recurrent spontaneous abortion. The confirmation and recognition of this phenomenon have been established. We have also covered this model adequately in our article.

In addition, other models are not considered to simulate the mechanism of recurrent miscarriage well. And there is a lack of comprehensive research on other models of recurrent abortion. Therefore, in order to ensure rigor and control heterogeneity, we have selected this model as an inclusion and exclusion criteria over other animal models. I am not sure if my expression can make you understand. If you still feel that it should be include more data on mice breed after reading my explanation, I am willing to respect your suggestion.

② Information of the studies in our article: The limitations of the included articles.

The comprehensiveness and accuracy of the information extracted in this study were ensured through a series of trade-offs. Apart from the administration routes, we also present details such as the source of MSC, dosage for each study, the efficacy of MSC in treating URSA may be influenced by these interventions. The information can be found in the table provided on pages 11-12. Furthermore, We proactively contacted authors whose articles were unreported or had missing data, but don't always get a positive response. The limited information available in the existing literature makes it difficult to conduct meta or subgroup analyses of other factors. This shortcoming was also mentioned in our discussion part. Moving forward, we will continue to closely monitor research in this field and eagerly anticipate discovering additional high-quality animal experiments for further exploration.

2.Are Mesenchymal Stromal Cell therapies FDA approved?

Response：

Thank you for inquiring. Currently, there are 10 globally approved mesenchymal stem cell products. However, the FDA has yet to grant approval for any mesenchmal stem cell products for market distribution. We learned that the investigational drug Ryoncil, developed by Mesoblast, is a mesenchymal stem cell product that has finished the Phase III trials and submitted an new drug application (NDA) application, which was anticipated to be the inaugural mesenchymal stem cell product in FDA approval. On August 4, 2023, however, the FDA refused to approve it. Mesoblast was requested to provide additional data in order to bolster the application. We are of the opinion that, going forward, stem cell therapies are extremely promising for FDA approval as clinical evidence becomes more abundant.

3.What are the long and short term side effects of MSC treatment?

Response：

Thank you for your insightful and probing inquiries. MSC treatment was shown to be safe due to their low immunogenicity. However, the differentiation potential and ability to promote tumor growth still provide concerns for MSCs clinical use. There are also some limitations in the use of stem cells for cell therapy such as finite replicative lifespan, ethical consideration, and the probability of somatic mutation.

4.What are the other diseases in which MSCs therapies are of importance.

Response：

The question you posed is greatly appreciated. Generally, MSCs therapies are thought to be capable of a wide range of applications, especially in autoimmune and graft-versus-host disease and various regenerative diseases.MSCs exhibit remarkable immunomodulatory and repairing capacity. MSCs can migrate to damaged tissue and directly differentiate into defective cells for repair, secrete many key trophic factors, and inhibit T, B and NK cell proliferation to suppress the inflammatory response, thus involved in tissue repair and regeneration as well as preventing immune rejection .

Reviewer #2: 

Suggestions for Improvement:

Clarity in Introduction: The Introduction section could benefit from enhanced clarity and conciseness. Providing a global context for the significance of URSA research and the broader relevance of MSC therapy would make the introduction more reader-friendly.

Simplify sentence structures and increase conciseness for better readability.

Incorporate a brief paragraph highlighting the global significance of URSA, particularly its impact on women's health and reproductive outcomes.

Expand on the importance of URSA research, emphasizing its relevance not only in clinical contexts but also within the broader fields of reproductive medicine and immunology.

Response：

Clarity in Introduction: We sincerely thank the reviewer for careful reading. As suggested by the reviewer, to enhanced clarity and conciseness, we revised some expressions in the introduction section, section 2.3 and 3.3. These changes will not influence the content and framework of the paper. In addition, we also explained the importance of studying URSA in brief words ( line 56-60). And you can see our modification traces when you browse in revision mode. We appreciate your warm work earnestly and hope that the correction will meet with approval.

Hypothesis and Objectives: Explicitly stating the hypothesis or main research question and outlining specific study objectives would provide readers with a clearer roadmap of the study's goals.

Materials and Methods: Clarify the rationale for specific eligibility criteria, such as the choice of CBA/J and DBA/2 mice as the animal model, to justify their selection. Provide a brief explanation of why the chosen primary and secondary outcome measures were selected and how they relate to the research question. 

Response：

Thank you for this suggestion. 

Clarify the rationale for specific eligibility criteria:The abortion-prone model of CBA/J × DBA/2 mating was used for recurrent abortion research since 1994, which are a well-investigated model of immunologically mediated recurrent pregnancy failure. Current studies suggest that the spontaneous abortion in this model was identified to relate to systemic maternal immune inflammation, increased complement deposition, and overactivation of NK cells and T cells in the maternal–fetal interface, which bears close resemblance to human recurrent miscarriage. The confirmation and recognition of this phenomenon have been established. We have also covered this model adequately in our article, as shown in line 349-359. .

In addition, other models are not considered to simulate the mechanism of recurrent miscarriage well. And there is a lack of comprehensive research on other models of recurrent abortion. Therefore, in order to ensure rigor and control heterogeneity, we have selected this model as an inclusion and exclusion criteria over other animal models. 

A brief explanation: This study aims to evaluate the pre-clinical evidence for the MSC treatment for URSA. To evaluate the efficacy, we chosen the primary outcome as the embryo absorption rate, and the secondary outcome as cytokine expressions (such as the level of IL4, IL10, IFN-γ, and TNF-α). The absorption rate of embryos is the most straightforward indicator for evaluating the impact of MCS treatment on mouse pregnancy outcomes, thus we have designated it as primary outcome measure. The secondary outcome markers were defined as cytokines, which were determined based on the mechanism of action of MCS and the pathogenesis of URSA.

Interpretation of Findings: Adding interpretation or discussion of the significance of the results, especially in relation to URSA treatment, would enhance the clarity of the Results section. Organize the presentation of data by introducing subheadings for each outcome measure (e.g., embryo absorption rate, cytokine levels) to enhance readability. After presenting the data for each outcome, include an interpretation or discussion of the significance of these findings in the context of URSA treatment.

Thank you for your valuable advice. In this study, embryo absorption rate is our primary outcome and inflammatory factors are secondary. We have therefore included subheadings for "primary outcome" and "secondary outcome" to enhance readability. In additon, we have includes an interpretation of each outcome indicator in the result parts (page 15-20, lin 220-222. line 246-248, line 273-274, line 302-304), and expand on the significance of these findings in the context of URSA treatment (page 16-21, lin 238-239, line 264-265, line 292-293, line 311-312). Furthermore, we have highlighted the implications of these findings for URSA treatment in the discussion section (page23-24,line 360-400).

Quantification of Evidence: Including specific quantitative results or effect sizes from the meta-analysis in the conclusion would emphasize the magnitude of observed effects.

Consider including specific quantitative results or effect sizes from the meta-analysis in the conclusion to underscore the magnitude of observed effects.

Conclude with a forward-looking statement about the potential impact of this research on URSA treatment and women's reproductive health to leave a strong impression on readers.

Forward-Looking Statement: Concluding with a forward-looking statement regarding the potential impact of this research on URSA treatment and reproductive health would leave a strong impression on readers.

The suggestion you have provided is greatly appreciated. We have summarized the results of the meta-analysis in the first paragraph of the discussion (page 22, line 325-336), and also have included additional comprehensive descriptions of MSCS treatment for URSA. Considering that we have presented the specific quantitative results or effect sizes in detail in the part of result (page 15-21), we did not repeat the above statistics again in the discussion part., which contributes to providing readers with a comprehensive analysis of the underlying mechanisms and future prospects, thereby enhancing the overall coherence of this article. In addition, at your suggestion, we have conclude with a forward-looking statement about the potential impact of this research on URSA treatment and women's reproductive health to leave a strong impression on readers (page 26-27, line 464-471).

Sincerely,

Dr. MA

---

## [Editor Report · Decision Letter 1]

9 Nov 2023

Efficacy of Mesenchymal Stromal Cells in the Treatment of Unexplained Recurrent Spontaneous Abortion in Mice: an Analytical and Systematic Review of Meta-analyses

PONE-D-23-25070R1

Dear Dr. MA,

We’re pleased to inform you that your manuscript has been judged scientifically suitable for publication and will be formally accepted for publication once it meets all outstanding technical requirements.

Kind regards,

Syed M. Faisal, Ph.D.

Academic Editor

PLOS ONE
---

## [Editor Report · Acceptance letter]

15 Nov 2023

PONE-D-23-25070R1 

Efficacy of Mesenchymal Stromal Cells in the Treatment of Unexplained Recurrent Spontaneous Abortion in Mice: an Analytical and Systematic Review of Meta-analyses 

Dear Dr. Ma:

I'm pleased to inform you that your manuscript has been deemed suitable for publication in PLOS ONE. Congratulations! Your manuscript is now with our production department. 

Kind regards, 

on behalf of

Dr. Syed M. Faisal 

Academic Editor

PLOS ONE